# Morphological and Motor Ability Adaptations Following a Short-Term Moderate-Intensity Strength Training Intervention in a Sedentary Adult Male with Asymmetrical Bilateral Spastic Cerebral Palsy: A Case Study

**DOI:** 10.3390/jfmk10040442

**Published:** 2025-11-17

**Authors:** Aleksandra Popović, Marko Kapeleti, Igor Zlatović, Milica Jankucić, Anastasija Kocić, Vladimir Mrdaković, Marija Macura

**Affiliations:** Faculty of Sport and Physical Education, University of Belgrade, Blagoja Parovića 156, 11030 Belgrade, Serbia; marko.kapeleti@fsfv.bg.ac.rs (M.K.); igor.zlatovic@fsfv.bg.ac.rs (I.Z.); milicajankucic@gmail.com (M.J.); anastasija.kocic@fsfv.bg.ac.rs (A.K.); vladimir.mrdakovic@fsfv.bg.ac.rs (V.M.); marija.macura@fsfv.bg.ac.rs (M.M.)

**Keywords:** movement disorders, muscle weakness, neurology, rehabilitation, resistance

## Abstract

**Background**: Cerebral palsy (CP) is a group of permanent disorders affecting movement, posture, and balance. Spasticity is the most common movement disorder in CP, and muscle weakness is its primary impairment. There is a lack of studies that have examined the effects of short-term, moderate-intensity strength training (ST) in adults with CP, whereas recommendations suggest that long-term interventions are necessary for substantial improvements in strength in the CP population. This study investigated the effects of a 5-week, moderate-intensity ST intervention, that targets various upper and lower extremity muscles, on multiple morphological characteristics (MC) and motor abilities (MA) in a sedentary 30-year-old adult male with asymmetrical bilateral spastic CP level II. **Methods**: Body composition, maximal knee strength, maximal squat strength, leg explosive strength, and hip mobility were assessed before and after the ST intervention. **Results**: Changes in body composition were modest (0.6–6.4%). Maximal knee strength increased moderately on the less spastic side (40.7–65.9%) and substantially on the more spastic side (118.5–130.6%). Hip mobility showed a similar pattern, with small to moderate improvements (11.4–30.0%), while maximal squat strength and leg explosive strength increased moderately (29.5–46.3%). **Conclusions**: A short-term, moderate-intensity ST intervention produced meaningful improvements in MC and MA in this subject, especially on the more spastic side. The applied ST program was feasible and potentially efficient, and the results of this single-case study support its approach and methodology in necessary future studies on larger trials in an attempt to generalize these preliminary findings. This in turn may encourage practitioners to promote increased participation in physical activity among individuals with CP, given the short-term period of adaptations. The study discusses the potential of further refinement of the existing CP-specific ST guidelines and load programming aspects.

## 1. Introduction

Cerebral palsy (CP) is a group of permanent disorders affecting movement, posture, and balance [1]. It is characterized by atypical motor control, resulting from non-progressive disturbances in the developing fetal or infant brain [2]. CP is the most common childhood physical disability, occurring in approximately 2.5 per 1000 live births. Nearly 80% of cases are idiopathic and occur within the prenatal period [3]. The most common CP subtype is bilateral spastic, with prevalence of 54.9% and 1.16 per 1000 live births [4]. The number of adults with CP is increasing, and they experience physical and mental health difficulties, leading to lower quality of life and increased risk of premature mortality, often from undiagnosed, preventable, noncommunicable diseases [5,6].

Spasticity is the most common movement disorder in CP [1], and muscle weakness is its primary impairment [7]. Adults with CP commonly show reduced muscle strength and endurance, and they need to maintain higher physical fitness levels than the general population to offset the decline in function associated with aging and the condition [8]. Therefore, promoting increased participation in physical activity is essential, as such engagement can gradually replace therapies that played a central role during childhood and adolescence, especially during the transition into adulthood [9]. Evidence shows that strength training (ST) is not linked to an increase in spasticity in children with CP, suggesting that ST is not contraindicated, but that it is beneficial for both upper and lower extremity training [9,10]. Since physical activity and exercise guidelines for the general population do not include specific suggestions for people with CP, Verschuren et al. (2016) published the first evidence-based CP-specific ST guidelines derived from a comprehensive review and analysis of the literature, expert opinion, and extensive clinical experience [9].

Verschuren et al. (2016) suggested in their guidelines that long-term training interventions of at least 12–16 consecutive weeks with progressive intensities may be needed to experience substantial improvements in strength in people with CP [9]. Regarding other specific guidelines, it was also suggested 2–4 sessions per week on non-consecutive days, 1–3 sets of resistance exercises at 6–15 repetitions of 50–85% of one-repetition maximum, which applies to both upper and lower extremities. Regarding the type of exercises and resistance, progression should be made from primarily single-joint, machine-based resistance exercises to machine plus free-weight, multi-joint and closed-kinetic chain resistance exercises [9]. According to these recommendations and lack of studies that examined the effects of short-term, moderate-intensity ST in adults with CP, the question arises as to whether the short-term period of a relatively reduced training load is sufficient to produce clinically significant ST benefits in this population. This could result in an increased promotion of physical activity by the practitioners and consequently increased participation of adult CP population because of a positive impact and attitude toward this level of physical activity.

This study aims to investigate the effects of a 5-week, moderate-intensity ST intervention on morphological characteristics (MC) and motor abilities (MA) in a sedentary adult male with asymmetrical bilateral spastic CP. We hypothesized that this ST intervention is potentially sufficient to produce significant morphological and motor ability adaptations in an adult male with CP, likely through neural adaptations, given the short-term window of intervention [11]. The study presents approach, methodology, and preliminary single-case results as a basis for necessary future adult CP case series and randomized controlled trials. The study discusses potential further refinement of the existing CP-specific ST guidelines and load programming aspects. The study addresses the health aspects of exercise science with direct applications to neurology and rehabilitation, thereby contributing to improved patient management and the advancement of knowledge across exercise and neurological sciences.

## 2. Case Description

This study examined a 30-year-old male with CP (ICD-10 code: G80, Bilateral Spastic, Left-Side Dominant). The subject walks independently with some limitations (level II according to the Gross Motor Function Classification System) and has no prior experience in organized physical activity. The subject’s intellectual functioning is well preserved.

MC and MA were assessed before and after the ST intervention (Figure 1). Body composition was measured using a bioimpedance analyzer (InBody720, Biospace Co., Seoul, Republic of Korea). Maximal knee strength was assessed with an isometric dynamometer (Kin-Kom AP125, Chatex Corp., Chattanooga, TN, USA). The subject performed three unilateral isometric knee flexion and extension contractions at 60° of knee flexion [12]. The rest between repetitions was 2 min. Maximal squat strength was assessed with a half-squat on a Smith machine. The external load increased until the subject could no longer complete a repetition successfully. The rest between attempts was 2 min. Leg explosive strength was assessed with three attempts of squat jumps and countermovement jumps on a force platform (AMTI, Watertown, MA, USA; Model BP600400). The rest between attempts was 1 min. Hamstrings flexibility was assessed with passive straight leg raise in a supine position, with measured outer leg angle. Hip adductors flexibility was assessed with passive hip abduction (straddle split) in a standing position, with measured distance between the medial borders of the feet. Data were obtained through the valid and reliable movement analysis program Kinovea (v.0.9.5-x64, https://www.kinovea.org/, accessed on 20 October 2025) [13]. The subject was familiarized with several practice trials on the testing day. The repetition with the best result was used for further analysis.

The training intervention lasted 5 consecutive weeks, with 3 sessions per week on non-consecutive days. Each session included 2–3 circuit sets of 8–12 exercises targeting different upper- and lower-body muscle groups (1–2 exercises per muscle group), single- and multi-joint, bilateral, open-kinetic chain, machine-based movements (Table 1), with 60-s rests between exercises and 1–3 min between sets. Closed-kinetic chain and free-weight exercises were excluded due to the subject’s inexperience and the short duration of the program. The number of repetitions ranged from 10 to 15, at the rating of perceived exertion of 4–6 out of 10 (somewhat hard to hard), which roughly corresponded to 40–60% of one-repetition maximum in this subject [14], depending on the exercise and training week. The subject did not do every set to failure. Exercise intensity increased, while the number of repetitions decreased during the first three weeks, and vice versa during the last two weeks. The warm-up included 10 min of low-intensity cycling on an ergometer and 5–10 min of dynamic mobility. The cooldown included 3 consecutive repetitions of proprioceptive neuromuscular facilitation stretching for the hamstrings and hip adductors, consisting of 8-s contractions (isometric, concentric, or eccentric), situated between 15-s passive stretching. The subject carried out the intervention under the guidance and supervision of the physical exercise professional.

### Results

The effects of the applied ST intervention are shown in Table 2 and Figure 2.

Changes in body composition were modest (0.6–6.4%) (Table 2). Maximal knee strength increased substantially on the left (118.5–130.6%) and moderately on the right side (40.7–65.9%) (Figure 2A). A similar pattern was observed for hip mobility, with small to moderate improvements (11.4–30.0%) (Figure 2D). Increases in maximal squat strength and leg explosive strength highlight moderate enhancements (29.5–46.3%) (Figure 2B,C).

## 3. Discussion

The goal of this study was to investigate the effects of a short-term, moderate-intensity ST intervention on MC and MA in a sedentary adult male with asymmetrical bilateral spastic CP level II. The results supported the hypothesis, showing that a 5-week moderate-intensity ST intervention likely produced meaningful improvements in MC and MA, particularly on the more spastic side. Given the short-term window of adaptations and the fact that all MC variables showed trends in the desired direction of adaptation (Table 2), even these modest changes can be considered clinically important, especially when considering the population to which the subject belongs. If generalized in future larger trials, these findings may encourage practitioners to promote increased participation in physical activity among individuals with CP, which may result in a positive impact and attitude toward physical activity of this population.

The unequal improvement of the left and right sides in maximal knee strength and hip mobility reduced initial bilateral imbalances (Figure 2A,D), which is likely explained by gradual habituation to greater muscular exertion and differences in neural adaptation between the initially asymmetrical sides, as shown in a related CP case study [15]. These uneven adaptations and the tendency toward balance were possibly due to differences in the initial spasticity levels between body sides, given that the exercises were bilaterally targeting the left and right side to the same extent in terms of applied training load. The more spastic side of the body (left) represents the more severely affected entity, whereas the less spastic side (right) reflects a milder manifestation of CP, suggesting that large gains may be attainable on the more affected side even with a relatively reduced training load over the short term. This finding highlights the necessity to incorporate this notion into the load programming aspects of CP-specific interventions, when initial spasticity and functional levels differ between the body sides. The improvements in maximal squat strength and leg explosive strength reflect improvements in overall lower-body function, clinically important for this population (Figure 2B,C).

Only two previous studies were found that were considered similar in part to the present study in terms of short-term effects (<8 weeks) of ST intervention in adults with CP. The present study is in line with a previous single-case study that showed positive adaptations in muscle activation and muscle strength after 4 weeks of isokinetic ST intervention in a sedentary adult with CP [15], supporting the hypothesis that neural adaptations likely played a primary role in the neuromuscular improvements, possibly through timing and homeostatic mechanisms of neuroplasticity—such as enhanced synaptic efficacy, synaptic scaling, changes in synaptic strength, and modulation of neuronal excitability [16]. On the other hand, an earlier study in a sample of 11 sedentary adults with CP showed no significant changes in leg press and chest press one-repetition maximum after 3 weeks of baseline (introductory) training with minimum intensity intended primarily for learning the exercises for future long-term progressive ST intervention [17]. As in the present study, the previous two studies show that adults with CP can increase strength over the short term, but only with sufficient training intensity, which altogether highlights the importance of tailored ST guidelines for this population.

In contrast to the previous studies, this study is the first one to examine specifically the effects of short-term, moderate-intensity ST intervention that targets various upper and lower extremity muscles and monitors multiple MC and MA in an adult with CP. The training program was carried out smoothly, the subject did not report any discomfort other than fatigue induced by exercise and the results of the intervention were clinically significant, which all highlight its potential feasibility and efficiency. There are a couple of limitations related to the single-subject design that should be considered and questioned in future studies. Firstly, improvements may partly reflect lower motivation and limited effort in the pre-test or learning effects during the program, related to the subject’s condition and inexperience. Secondly, this subject may not represent the broader adult CP population, since his general level of impairment is mild and intellectual functioning is well preserved. Lastly, even though the experimental protocol incorporated familiarization and multiple repetitions in every test, it did not account for natural variability and measurement error during initial and final testing by showing statistical measures of central tendency and dispersion. Due to all of these limitations, the present study remains preliminary in its scope, results and interpretation, and future studies should focus on exploring the real magnitudes of these effects.

As this single-case study applied relatively reduced training load over reduced duration of the program relative to the recommendations for ST interventions in people with CP [9], future studies should replicate this methodology and ST program across diverse adult CP cases and in randomized controlled trials compared to healthy controls to assess the generalizability of these preliminary results and potentially further contribute to the refinement of the existing CP-specific ST guidelines. It is recommended to incorporate in future studies also electromyographic and spasticity-specific metrics (Modified Ashworth Scale or Modified Tardieu Scale) in order to explain neuromuscular improvements more thoroughly. The approach of distinguishing between sides differently affected by the condition in people with CP should also be incorporated into the further refinement of load programming aspects of ST interventions in people with CP, perhaps by introducing unilateral exercises, different external loads, and separate rating of perceived exertion assessments.

## 4. Conclusions

A short-term, moderate-intensity ST intervention likely produced meaningful MC and MA adaptations in a sedentary adult male with asymmetrical bilateral spastic CP level II, suggesting that this population may benefit from moderate ST even over brief periods. Maximal knee strength and hip mobility adapted differently depending on the body side, which is in accordance with the differences in spasticity levels between sides in this subject. These spasticity levels represent different severities of CP, which should be observed and treated as different entities due to different adaptations to the same stimulus. These initial differences should also be taken into account for further refinement of the existing CP-specific ST guidelines and load programming aspects, perhaps by introducing unilateral exercises, different external loads, and separate rating of perceived exertion assessments. The applied ST program was feasible and potentially efficient, and the results of this single-case study support its approach and methodology in necessary future adult CP case series and randomized controlled trials compared to healthy controls in an attempt to generalize these preliminary findings.

## Figures and Tables

**Figure 1 jfmk-10-00442-f001:**
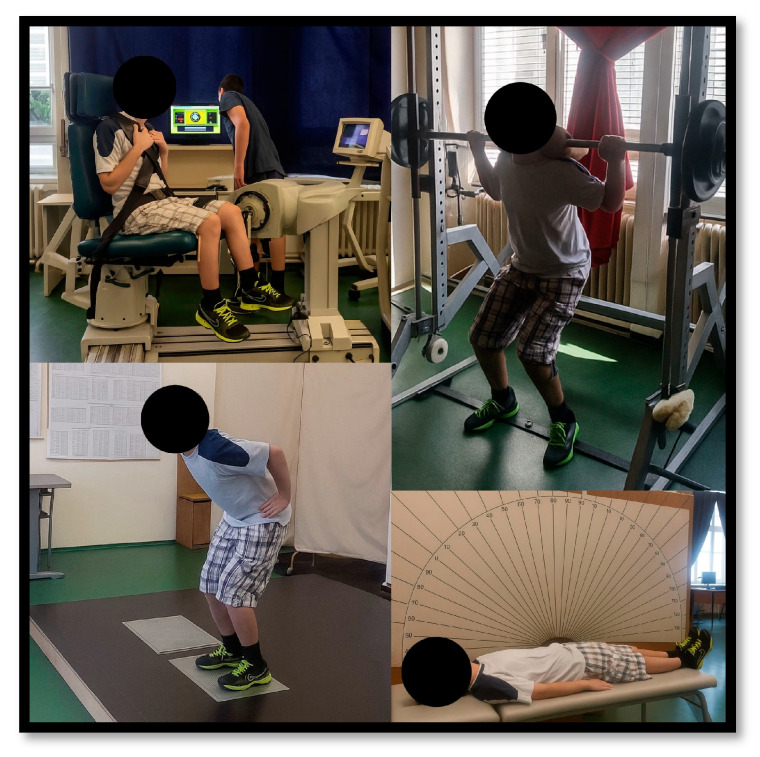
Subject performing the tests.

**Figure 2 jfmk-10-00442-f002:**
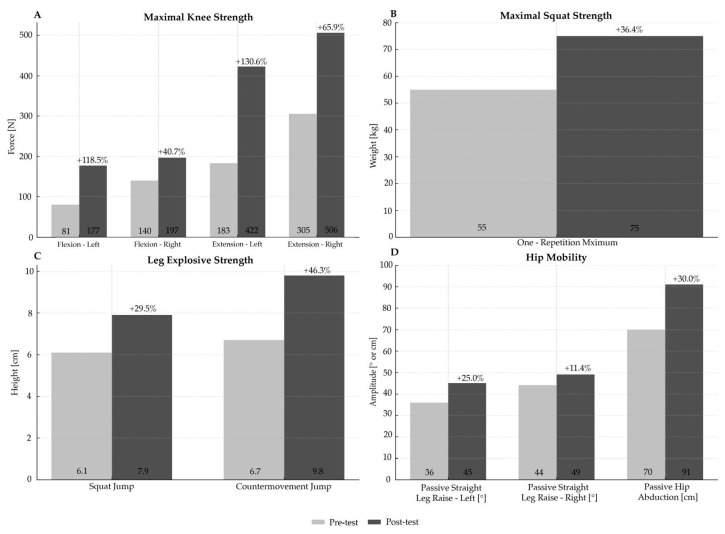
Strength and mobility results following a 5-week, moderate-intensity ST intervention in a 30-year-old sedentary male with asymmetrical bilateral spastic CP level II. (**A**)—maximal knee strength; (**B**)—maximal squat strength; (**C**)—leg explosive strength; (**D**)—hip mobility.

**Table 1 jfmk-10-00442-t001:** The example of exercises used in the ST program.

Exercise Name	Exercise Description
Leg press	While sitting, push the resistance platform with both legs to full knee extension, then slowly lower the resistance platform down till comfortable hip and knee flexion.
Knee extension	In seated position with adjusted resistance bar over anterior shin, extend both knees till full extension, then slowly lower the resistance bar.
Knee flexion	In prone position with adjusted resistance bar over posterior shin, flex both knees till full flexion, then slowly lower the resistance bar.
Hip abduction	While sitting, push thigh pads laterally till full hip abduction, then slowly return to the starting position.
Hip adduction	While sitting, pull thigh pads medially till full hip adduction, then slowly return to the starting position.
Chest press	In supine position, push the resistance handles forward to full elbow extension, then slowly return to the starting position.
Shoulder press	While sitting, push the resistance handles overhead, to full elbow extension, then slowly return to the starting position.
Lat pull down	While sitting, pull the resistance handles as close to the chest as possible (pronated grip), then slowly release to the starting position.
Seated row	While sitting, pull the resistance handles as close to the torso as possible (parallel grip), then slowly release to the starting position.
Abdominals	While in supine position with knees bent and stabilized, perform trunk flexion till the scapulas are above the floor, then slowly return to the floor.
Triceps pull down	While standing, extend the elbows downward against cable resistance till full extension, then slowly return to the starting position.
Biceps curl	While sitting on Scott bench, flex the elbows upward against resistance bar till full flexion, then slowly return to the starting position.

**Table 2 jfmk-10-00442-t002:** Body composition results following a 5-week, moderate-intensity ST intervention in a 30-year-old sedentary male with asymmetrical bilateral spastic CP level II.

Tests	Variables	Pre-Test	Post-Test	%
Body composition	Body Mass [kg]	65.4	66.3	+1.4
	Total Body Water [L]	35.9	37.0	+3.1
	Skeletal Muscle Mass [kg]	27.3	28.0	+2.6
	Body Fat Mass [kg]	16.5	16.1	−2.4
	Body Fat Percentage [%]	25.3	24.2	−4.3
	Visceral Fat Area [cm^2^]	88.8	83.1	−6.4
	Protein [kg]	9.7	9.9	+2.1
	Mineral [kg]	3.28	3.30	+0.6

## Data Availability

The original contributions presented in this study are included in the article. Further inquiries can be directed to the corresponding author.

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
