# Peer review of "J. Funct. Morphol. Kinesiol.2025, 10(4), 442;https://doi.org/10.3390/jfmk10040442"

_jfmk, 2025, doi:10.3390/jfmk10040442_

Round 1

Reviewer 1 Report

Comments and Suggestions for Authors

The manuscript presents a single-case experimental design investigating short-term strength-training effects in an adult with spastic cerebral palsy (CP). Overall, the study is clearly written, ethically compliant, and methodologically transparent, but it remains preliminary in scope and interpretation, with several limitations that constrain its generalizability and scientific strength.

From a scientific perspective, the paper addresses an underexplored niche—the feasibility and potential benefits of moderate-intensity resistance training in adults with CP—an area where most existing evidence pertains to pediatric populations. The rationale is well introduced, grounded in the lack of adult-specific strength-training studies and the gap in CP-specific exercise guidelines. The authors appropriately cite foundational literature, particularly Verschuren et al. (2016), to justify their hypothesis. The theoretical framing—that short-term neural adaptations may precede structural hypertrophy in this population—is sound and consistent with neuromuscular physiology. Nonetheless, the narrative would benefit from a deeper mechanistic discussion of neural plasticity in CP and its responsiveness to resistance training.

Methodologically, the study is carefully executed for a case report. The description of participant characteristics, measurement tools, and training protocol is detailed and replicable. The use of validated instruments (InBody720 for body composition, Kin-Kom dynamometer for knee strength, and AMTI force platform for explosive strength) supports internal validity. The training prescription is appropriately cautious and well justified for a previously sedentary adult with spasticity. However, as a single-subject design without repeated baseline measures or post-intervention follow-up, the study lacks internal control for natural variability, motivation, or measurement error. Reporting confidence intervals or effect magnitude interpretations would strengthen the quantitative presentation even in a case format. Moreover, the reliance solely on pre- and post-test comparisons precludes inferences about causality or temporal adaptation trajectories.

The results are presented clearly, with modest improvements in body composition and substantial increases in strength and mobility—particularly on the more affected side. While plausible and encouraging, the magnitude of change in maximal knee strength (up to 130%) seems unusually high for a five-week intervention, raising concerns about learning effects, submaximal effort during baseline testing, or measurement sensitivity. A discussion acknowledging potential test–retest learning or motivation effects would enhance scientific rigor. Similarly, no electromyographic or spasticity-specific metrics (e.g., MAS or Tardieu Scale) were collected, which would have helped contextualize motor improvements and differentiate neural from muscular adaptations.

The discussion interprets the findings appropriately, relating them to the limited prior studies in adults with CP and reinforcing the potential for positive short-term adaptations. The emphasis on side-to-side asymmetry and differential adaptation is an insightful and clinically meaningful observation that could inform individualized load prescription. However, the claim that results were “meaningful” or “positive” should be tempered given the single-case design and absence of statistical or functional benchmarks (e.g., minimal clinically important difference). The section could also benefit from a more critical reflection on external validity, emphasizing that this participant’s relatively mild impairment and high cognitive functioning may not represent the broader CP adult population.

Ethical and procedural standards are properly met—the study was approved by an institutional ethics committee, conducted according to the Declaration of Helsinki, and followed CARE guidelines. The data availability and conflict-of-interest statements are appropriate. Figures and tables are clear, though the inclusion of effect size estimations or absolute performance values (e.g., in Newtons or kg for strength measures) would improve interpretability. The English language and structure are overall clear and professional, though minor grammatical polishing and condensation of repetitive sentences would enhance readability.

The manuscript offers a well-executed, valuable preliminary observation in a field where adult CP exercise research is scarce. Its methodological transparency and clinical relevance support its contribution as a case report or pilot framework. However, the scientific strength is limited by the single-subject design, absence of objective neuromuscular correlates, and potential measurement artifacts. The conclusions should therefore remain descriptive rather than inferential, emphasizing feasibility and hypothesis generation rather than efficacy.

Reviewer 2 Report

Comments and Suggestions for Authors

I have read with high interest this case study about the use of strength training in a subject with cerebral palsy. The topic is very interesting since the treatment of this condition is changing to a more functional philosophy. However, authors have not deepen into the functional assessment of the participant considering upper and lower limbs, they focused on a general picture of the case. They also must understand that the use o a single patients cannot allow to generalize the results observed. I suggest authors to perform the protocol on several more patients and to perform a real pilot trial with a minimum sample, as the present results are not useful to discuss the results with others.

Introduction is adequate, explaining the definition of CP and prevalence data. The use and safeness of using strength training in CP is also supported by citations. However, the aims of the study are too high to use a single case as sample. Authors presents this study as a “pilot study for future trials” but I cannot consider this a pilot study since a single case cannot guide any trial. I recommend to consider changes in the title to avoid “pilot study”.

Materials and methods: the participant physical condition description with the Gross Motor Function Classification could be acceptable, although more specific information will be desirable regarding lower and upper limb’s function and walking capability. The intellectual functioning is described as “above average”, without any objective data or specifying how this was assessed.

Lines 99-100, 107: citations about the validity of these instruments would be desirable, if possible.

Lines 102-103: some pictures about the machines and squat could help to understand the protocol.

Results are very well described with graphics visually supporting all the pre-post differences.

Discussion: the discussion is quite weak since the present study has only one participant and the result obtained differ from two studies with more participants. Authors do not highlight that the differences observed could be related with the bias of measuring a solely participant.

Line 181: please specify the mentioned “own limitations”.

Line 182: consider to add “a” between “previous” and “study”

A limitation section of this study is recommended at the end of the discussion, as most of the future studies recommendations are limitations of the present study.

Reviewer 3 Report

Comments and Suggestions for Authors

This is an interesting and well-written case study addressing an underexplored topic - the impact of a short-term, moderate-intensity strength training (ST) intervention on morphological and motor outcomes in an adult with cerebral palsy (CP). The manuscript is methodologically sound, clearly structured, and highly relevant for both clinical and exercise science audiences. It fills a gap in the literature by focusing on an adult CP population, for which training adaptations are rarely documented.

The introduction provides sufficient theoretical and clinical background, and the methodology is detailed enough to ensure reproducibility. The authors appropriately contextualize their findings within the current evidence base and highlight the relevance of neural adaptations in short-term interventions.

The paper provides valuable preliminary evidence supporting the potential benefits of moderate-intensity strength training in adults with CP. However, this contribution could be better emphasized by contrasting the present results with the limited number of previous adult CP case studies.

Round 2

Reviewer 2 Report

Comments and Suggestions for Authors

The authors have successfully solved all the issues proposed. The manuscript findings are interesting but limited due to the study design.